# Effective Connectivity Evaluation of Resting-State Brain Networks in Alzheimer’s Disease, Amnestic Mild Cognitive Impairment, and Normal Aging: An Exploratory Study

**DOI:** 10.3390/brainsci13020265

**Published:** 2023-02-04

**Authors:** Fatemeh Mohammadian, Maryam Noroozian, Arash Zare Sadeghi, Vahid Malekian, Azam Saffar, Mahsa Talebi, Hasan Hashemi, Hanieh Mobarak Salari, Fardin Samadi, Forough Sodaei, Hamidreza Saligheh Rad

**Affiliations:** 1Department of Medical Physics and Biomedical Engineering, Tehran University of Medical Sciences, Tehran 1417613151, Iran; 2Department of Psychiatry, Roozbeh Hospital, Tehran University of Medical Sciences, Tehran 13185/1741, Iran; 3Medical Physics Department, Iran University of Medical Sciences, Tehran 1449614535, Iran; 4Wellcome Centre for Human Neuroimaging, UCL Queen Square Institute of Neurology, University College London, London WC1N 3AR, UK; 5Department of Biostatistics, Faculty of Paramedical Sciences, Shahid Beheshti Medical University, Tehran 1971653313, Iran; 6Department of Radiology, Advanced Diagnostic and Interventional Radiology Research Center, Imam Khomeini Hospital, Tehran University of Medical Sciences, Tehran 1417743855, Iran; 7Quantitative Magnetic Resonance Imaging/Spectroscopy Group, Tehran University of Medical Sciences, Tehran 1416753955, Iran

**Keywords:** Alzheimer’s Disease, amnestic mild cognitive impairment, resting-state functional MRI, effective connectivity, spectral dynamic causal modeling

## Abstract

(1) Background: Alzheimer’s disease (AD) is a neurodegenerative disease with a high prevalence. Despite the cognitive tests to diagnose AD, there are pitfalls in early diagnosis. Brain deposition of pathological markers of AD can affect the direction and intensity of the signaling. The study of effective connectivity allows the evaluation of intensity flow and signaling pathways in functional regions, even in the early stage, known as amnestic mild cognitive impairment (aMCI). (2) Methods: 16 aMCI, 13 AD, and 14 normal subjects were scanned using resting-state fMRI and T1-weighted protocols. After data pre-processing, the signal of the predefined nodes was extracted, and spectral dynamic causal modeling analysis (spDCM) was constructed. Afterward, the mean and standard deviation of the Jacobin matrix of each subject describing effective connectivity was calculated and compared. (3) Results: The maps of effective connectivity in the brain networks of the three groups were different, and the direction and strength of the causal effect with the progression of the disease showed substantial changes. (4) Conclusions: Impaired information flow in the resting-state networks of the aMCI and AD groups was found versus normal groups. Effective connectivity can serve as a potential marker of Alzheimer’s pathophysiology, even in the early stages of the disease.

## 1. Introduction

Alzheimer’s disease (AD), one of the most common types of dementia, imposes a high socioeconomic burden and stress on caregivers [1,2]. This neurodegenerative disease can affect a person’s orientation, immediate memory, attention, calculation, language, perception, visual abilities, and executive functions [3]. The global burden disease (GBD) 2019 estimates that the number of dementia patients will increase from 57.4 million in 2019 to 152.8 million in 2050 [4]. Because the disease process begins years before the clinical diagnosis [5], it is essential to understand the mechanism of AD in the early stages, known as amnestic mild cognitive impairment (aMCI) [6]. Early diagnosis and anti-dementia treatment strategies can prevent widespread neuronal death and delay disease progression [5].

The most important pathological markers in AD are extracellular β-amyloid deposition and intracellular accumulation of neurofibrillary tangles [7]. Structural and functional studies have identified the central nodes of the brain as the earliest sites of disease occurrence [8]. There are several known features to identify AD: global atrophy and local atrophy (prominent atrophy of the medial temporal lobe) [9]; β-amyloid deposition in dysfunctional hub centers [10]; hypometabolism in the temporal, parietal, and posterior cingula [9]. Several studies have shown a decrement in volumetric markers [11,12], the presence of amyloid markers [12,13], and a decrease in the functional connectivity (FC) in brain nodes of AD and aMCI patients [2,14,15,16].

The diagnosis of brain dysfunction widely benefits from functional magnetic resonance imaging (fMRI) as it is a non-invasive imaging technique that provides high spatial resolution maps with high reproducibility. fMRI detects human cortical neural activity based on blood oxygen level-dependent (BOLD) contrast signals [17]. Statistical correlation of low-frequency oscillations of the BOLD signal time series at resting-state defines as FC [18,19]. Most fMRI studies have focused on abnormal FC in AD and aMCI [16,20,21,22]. Nevertheless, there is a limitation on how resting-state networks with FC disruption operate and how information processing and the neural-signaling flow change. FC only reflects the interactions between different brain areas [23], while interactions between brain areas are directional and, therefore, not entirely defined by functional (directionless) connectivity [24]. 

The study of effective connectivity (EC) or directional effects allows the evaluation of intensity flow and signaling pathways in functional regions, which aids in diagnosing and predicting therapeutic responses in neurological and mental disorders [24]. This can provide a good understanding of the patterns of interaction between different areas of the brain [23]. Deposition of pathologic markers of Alzheimer’s disease in the brain region can affect the direction of the signal path [25,26]. Different methods of signal pathway investigation in fMRI, such as Granger Causality Analysis (GCA) and dynamic causal modeling )DCM(, have shown altered EC in resting-state networks of AD patients compared to the normal participants [23,27,28]. DCM, one of the most widely used modeling frameworks for EC inference from fMRI data, is a generative model of latent neural states [29]. Various DCM methods were developed to infer directional effects in brain regions. Among them, spectral DCM (or spDCM) [24] is an efficient biological model that explains and predicts BOLD signals caused by endogenous neural oscillations [30]. spDCM is used to model the intrinsic dynamics of a resting-state network and to infer the interaction between resting-state latent neural states [31]. This DCM estimates the directional connectivity in coupled populations of neurons [31] and offers a way to understand interregional interactions between predefined brain areas in resting-state networks. Liu Z et al. 2012 performed GCA to evaluate the EC among the resting-state networks. They found altered causal interactions in the resting-state network of AD patients (weaker interactions in the default mode network (DMN) and more robust causal connectivity in the memory network and executive control network), and they inferred that these findings may help in determining the neurophysiological mechanisms [32]. Chand et. al 2017, in a resting-state DCM study, reported aberrancies in the triple network’s (salience network (SN), executive control network, and DMN) interaction in MCI compared to the normal group, and they attributed these disorders to cognitive impairment [33]. The SN, involved in autonomic and homeostatic processing, has shown increased connectivity in AD patients [32,34]. In these patients, decreased connectivity within the DMN were associated with higher connectivity strength in the SN, possibly due to compensatory mechanisms [35,36]. A study by Scherr M et al. 2021 showed EC disruption in the DMN of AD patients, and considering the effect of the neuropathologic marker on the signaling pathway, they suggested testing molecular theories about the downstream and upstream spread of neuropathology in AD [25]. Heterogeneous and reduction-dominated alterations in EC are consistent with widespread reduction and bidirectional changes in FC in MCI and AD [29,37,38,39]. Despite the studies that have been conducted on EC in the resting state networks, the number of these studies is few, and most of them have focused on the DMN [23,25,40,41]. Therefore, changes in EC patterns and signaling hierarchy in the resting state networks in aMCI and AD patients are unknown to a large extent.

In this exploratory study, we hypothesize that examining the pattern of EC in different brain regions is effective for differentiating normal, aMCI, and late-onset AD. Therefore, we evaluated the signal pathways of resting-state brain networks to examine how the brain areas in the resting-state networks interact with each other in these three groups and how Alzheimer affects these patterns of interaction and alteration in the neural brain network signaling map. 

## 2. Materials and Methods

### 2.1. Demographic and Clinical Characteristics

The Tehran University of Medical Sciences ethics committee approved this study (IR.TUMS.MEDICINE.REC.1398.617). Detailed information about the imaging procedure was given to the participants and their authorized caregivers. The subjects signed an informed consent form approved by the Research Ethics Working Group. Through clinical evaluations and neuropsychological tests, three groups, including 14 normal (6 male -8 female), 16 aMCI (6 male -10 female), and 13 late-onset AD subjects (5 male -8 female), were selected by Yadman Institute for Brain, Cognition and Memory Studies (Tehran, Iran). Patient selection was based on the Mini-Mental State Examination score (MMSE), Montreal Cognitive Assessment score (MoCA), and Functional Assessment Staging score (FAST) (Table 1). 

#### 2.1.1. Inclusion Criteria

Patients were screened through a questionnaire, a clinical examination, and their history, and inclusion and exclusion criteria were defined. Inclusion criteria for normal aging (control group): Not suffering from neurological and psychological disorders such as depression or epilepsy. No visual or hearing impairments. No history of stroke or focal lesion in brain structural MRI imaging. No memory loss. Normal score in MMSE, MoCA, and FAST tests. Inclusion criteria in the aMCI group: Memory impairment confirmed by the patient’s family. There should be no disturbance in the daily activities and independence of the person. No dementia. Hippocampal atrophy confirmed by structural MRI. Medium score in MMSE, MoCA, and FAST tests. Inclusion criteria for the AD group: They were diagnosed based on clinical and neuropsychiatric evaluation and FAST, MMSE, and MoCA test scores.

In all three groups, people who did not smoke and did not drink alcohol were included in the study.

#### 2.1.2. Exclusion Criteria

For all groups, exclusion criteria were the following: mental and neurological diseases; high blood pressure; diabetes; alcohol dependence; substance abuse; history of head trauma; high signal lesions with a diameter > 5 mm or more than four lesions with a diameter < 5 mm using T2-FLAIR; other types of dementia such as vascular or mixed dementia, dementia with Lewy bodies, and frontotemporal dementia (FTD); diseases that may cause memory loss such as Parkinson’s disease, anemia, cancer, and other severe physical illnesses. Moreover, claustrophobia or lack of cooperation of the subject in immobility during and after imaging led to subject exclusion. 

### 2.2. MRI Acquisition

T1-weighted MPRAGE and rs-fMRI data were acquired using a 3.0 T MAGNETOM Prisma MRI scanner (Siemens Healthcare, Germany) at the National Brain Mapping Laboratory (Tehran, Iran). The scanning parameters of MPRAGE imaging were as follows: TR: 1840 ms; TE: 3.55 ms; TI: 800 ms; distance factor: 50%; the number of slices: 176; voxel size: 0.9 × 0.9 × 0.9 mm; flip angle: 7°; measurements: 1; multi-slice mode: sequential; series: ascending.

To acquire T2*-weighted (rs-fMRI) data, a single-shot echo-planar sequence was used with the following protocols: TR: 3000 ms; TE: 30 ms; distance factor: 0%; the number of slices: 45 per volume; voxel size: 2.8 × 2.8 × 3 mm; flip angle: 90°; measurements: 100, echo spacing: 0.5 ms; EPI factor: 80; multi-slice mode: interleaved; series: interleaved.

### 2.3. Pre-Processing

MNI normalization, tissue segmentation to gray matter, white matter, and CSF; realignment; co-registration; slice timing correction; segmentation; MNI normalization; and functional smoothing (Gaussian kernel of 4 mm) were performed using Statistic Parametric Mapping software (SPM) version 12 [31].

### 2.4. Resting-State Analysis 

#### 2.4.1. Independent Component Analyses (Group-ICA)

Group-ICA analysis was performed using Calhoun’s group-level ICA approach with CONN toolbox (www.nitrc.org/projects/conn, accessed on 24 December 2022, RRID: SCR_009550) [19] to extract resting-state networks. The fast-ICA algorithm was used for group-level independent component definition, and GICA3 back-projection was used for subject-level spatial map estimation. The “summary” option was selected in the ICA second-level results to explore the estimated ICA networks. At the tab of ICA tools, by choosing the “spatial match to template” option, the correlations between each group-level spatial map and CONN’s default network file were computed to identify networks of interest. The regions reported with high connectivity values on group ICA comparison were selected as sources: 32 ROI with coordinates from CONN atlas files (Appendix A)—MPFC: medial prefrontal cortex, LP(L): left lateral parietal, LP(R) right lateral parietal, PCC: posterior cingulate cortex, SML(L): left lateral sensorimotor, SML(R): right lateral sensorimotor, SMS: superior sensorimotor, VM: medial visual, VO: visual occipital, VL(L): left lateral visual, VL(R): right lateral visual, ACC: anterior cingulate cortex, Alnsula(L): left anterior insula, Alnsula(R): right anterior insula, RPFC(L): left rostral prefrontal cortex, RPFC(R): right rostral prefrontal cortex, SMG(L): left supramarginal gyrus, SMG(R): right supramarginal gyrus, FEF(L): left frontal eye field, FEF(R): right frontal eye field, IPS(L): left intraparietal sulcus, IPS(R): right intraparietal sulcus, LPFC(L): left lateral prefrontal cortex, PPC(L): left posterior parietal cortex, LPFC(R): right lateral prefrontal cortex, PPC(R): right posterior parietal cortex, IFG(L): left inferior frontal gyrus, IFG(R): right inferior frontal gyrus, pSTG(L): left posterior superior temporal gyrus, pSTG(R): right posterior superior temporal gyrus, CA: anterior cerebellar, CP: posterior cerebellar), represented eight resting-state networks: DMN, sensorimotor network (SMN), visual network (VN), SN, dorsal attention network (DAN), frontoparietal network (FPN), language network (LN), and cerebellar network (CN) [2,42].

#### 2.4.2. Spectral Dynamic Causal Modeling Analysis

EC analysis of resting-state networks was performed using SPM 12 and DCM 12.5. We used Spectral DCM analysis which estimates the time-invariant parameters of cross spectra. The original time series were replaced by their second-order statistics, i.e., cross spectra, and instead of assessing time-varying hidden states, we estimated their covariance, which does not change with time [31]. Linear regression was used to remove the main noise components of each voxel and each subject; a general linear model (GLM) containing movement parameter and confound time series from CSF and white matter (WM) was constructed and inverted. In addition, a GLM containing band pass frequency ranging from 0.01 to 0.1 Hertz (Hz) was modeled to focus on low-frequency fluctuations of the resting-state BOLD signal, removing any slow frequency drifts (< 0.01 Hz) and removing confounds with high frequency such as physiological noise: regressor for CSF signal, with spherical ROI with a radius of 6 mm and [0 −40 −5] coordinates on the center of the ROI and a regressor for the white matter signal, with a spherical ROI with a radius of 6 mm, and [0 −24 −33] coordinates on the center were defined and masked. The names and coordinates of the center of the 32 ROIs from 8 resting-state networks were defined and masked. Regional time series were extracted as the eigenvariate of all selected ROIs within an 8 mm radius around the targeted area, and DCM was constructed. Since, based on previous studies, the full model of spDCM is the best model at the group level [43,44,45], this model was selected in this study, and EC parameters were estimated, including bilateral and unilateral endogenous connectivity. Cross-spectral DCM (Full model with no exogenous inputs and non-stochastic effect) was constructed and inverted for each subject (Figure 1 and Figure 2: an example figure for one of the resting state networks, DMN). In estimated spDCM, the mean and standard deviation of the Jacobian matrix, which describes each resting-state network’s behavior—i.e., the EC—were calculated, and the findings in 3 groups were compared. Connections with strength below the 0.05 Hz threshold were excluded.

### 2.5. Statistical Analysis

THE IBM SPSS Statistics platform (version 26) was used to compare demographic and cognitive data variables. Normality of continuous variables checked with Kolmogorov–Smirnov test. Due to the non-normality of continuous variables, non-parametric tests were used to check the assumption of groups’ median equality and percentages. Kruskal–Wallis and Chi-square tests were employed for continuous and categorical variables, respectively. A pairwise comparison was performed using the Wilcoxon test. The predefined power was 0.8, and the level of statistical significance was considered *≤ 0.05*.

Causal interactions were investigated using spDCM in eight resting-state networks through Group-ICA analysis, and 32 ROIs were identified as sources for EC analysis to differentiate late-onset AD, aMCI, and normal. EC patterns of brain networks in the resting state were shown as directed graphs. The direction of the arrow indicates the direction of the causal effects, and its thickness indicates the strength of the connection. All spDCM assumptions were checked and approved. Confounding variables such as motion correction and realignment were considered in the computation process at the First Level analysis. The total observed power of all spDCM processes was 0.79 [46,47].

## 3. Results

### 3.1. Result of Demographic and Clinical Characteristics

In this study, the cohorts were not gender-equal (females more than men), but there were no significant differences between the groups. MMSE, MoCA, and FAST scores were significantly different between the three groups, and there was no significant difference in age between the three groups. The observed power of each main test is reported. Most of the observed powers were more than the predefined power (Table 1).

### 3.2. Results of spDCM Analysis 

#### 3.2.1. Default Mode Network

In the normal group, there is bilateral solid causal interaction between LP(L)⇋LP(R) (0.1357 Hz for LP(L)→LP(R) and 0.1519 Hz for LP(R)→LP(L) connection). Furthermore, in the normal group, connections with the origin and destination of LP(L) and LP(R) nodes are more robust than other connections. In aMCI and AD groups, there is almost no connection in LP(L)→LP(R), and there is a deficiency in the interaction from LP(R) to LP(L) connection. The MPFC→LP(L) connection with the frequency of 0.173 Hz in the normal group is almost disconnected in the aMCI and AD groups. Compared to the normal group, MPFC→PCC, LP(R)→MPFC, and PCC→LP(R) connections have become stronger in the aMCI group. The normal signaling from LP(R) to PCC (0.278 Hz) is almost interrupted in the aMCI group (Figure 3 and Appendix A). (Arrows indicate one-way and two-way causal interactions between nodes in resting-state networks.)

#### 3.2.2. Cerebellar Network

In the AD group, CA→CP connection and not vice versa has become stronger versus in the normal and aMCI groups (Figure 3 and Appendix A).

#### 3.2.3. Dorsal Attention Network

The normal strong bilateral causal interaction between FEFL⇋FEFR and normal signaling from IPS(R)→IPS(L) and IPS(R)→FEFR become weak in the aMCI group. There is almost no connection in IPS(R)→IPS(L) and IPS(R)→FEFR pathways in the AD group. In aMCI, new signaling has been formed with FEFR origin, IPS(R) and IPS(L) destination, and new signaling from IPS(L) to IPS(R) (Figure 3 and Appendix A).

#### 3.2.4. Sensorimotor Network

In the normal group, there is bi-directional causal interaction between SML(L) and SML(R). The SML(L)→SML(R) signaling is lost in the aMCI group and returns in the AD group (Figure 3 and Appendix A).

#### 3.2.5. Visual Network

The normal signal pathways from VM→VO (0.4184 HZ), VM→VL(L) (0.1847 HZ), VM→VL(R) (0.0966 HZ) become weak, and normal flows from VL(L) to VL(R) (0.1251 HZ) and from VL(L) to VO (0.091 HZ) become almost cut off in the AD group. In the aMCI and AD groups, a new signaling pathway from the visual lateral(R) to the visual lateral(L) is formed (Figure 3 and Appendix A).

#### 3.2.6. Salience Network

In aMCI and AD groups, the bilateral connections between Alnsula(L) and Alnsula(R), SMG(R)→SMG(L), and SMG(R)→Alnsula(R) connections become weak compared to the normal group, and RPFC(L)→RPFC(R) signaling is lost. Instead, new signal directions are formed, and these additional connections in the AD group become more; this network becomes complex in these two groups (Figure 3 and Appendix A).

#### 3.2.7. Language Network

In the normal group, there is a robust bilateral connection between PPC(L)⇋PPC(R) that becomes one-sided (PPC(R)→PPC(L)) in aMCI and AD groups. The PPC(R)→PPC(L) connection is weakened from normal to aMCI and AD. In AD and aMCI groups, LPFC(L)→PPC(L) connection is lost (Figure 3 and Appendix A).

#### 3.2.8. Frontoparietal Network

In the normal group, there is a robust bilateral connection between PPC(L) and PPC(R) that becomes one-sided in aMCI and AD groups, and the PPC(L)→PPC(R) connection is lost. In AD and aMCI groups, the LPFC(L)→PPC(L) connection is lost. The PPC)R(→PPC)L( connection is weakened from normal to aMCI and AD. See Figure 3 and Appendix A.

## 4. Discussion

We performed an EC-based analysis with the spDCM approach to investigate causal interactions in the resting-state network’s nodes of the normal, aMCI, and late-onset AD groups. Disruption of EC in the resting-state network of AD and aMCI can be due to hypometabolism and brain atrophy, β-amyloid deposition, and FC alteration. As our study showed, several studies also reported a decrease in EC in strength and quantity in AD compared to the normal group [25,27,32,33].

In DMN, the presence of bilateral causal interactions between LP(L)⇋LP(R) can be an indicator to differentiate the normal group from the AD group. Based on several studies, there is a higher inter-regional causal interaction between LIPC/RIPC and other nodes of DMN, so these nodes have a driving role in the DMN of the normal group [43,48,49]. The MPFC→LP(L) connection may be a good indicator for distinguishing the normal group from the other two groups. In both AD and aMCI groups, LP(R)→MPFC connection may provide the compensatory signaling to prevent network degradation. We found that the LP(R) node could be considered a hub in DMN due to the robust + and – causal interactions between this node and other nodes. Studies have shown that the stage of MCI can lead to abnormal connectivity within the DMN and other brain regions [32,50]. Compared to the normal group, higher frequencies in MPFC→PCC, LP(R)→MPFC, and PCC→LP(L) connections were found in the aMCI group, probably due to a compensatory mechanism, which could be an indicator for distinguishing these two groups. This new connection with the origin and destination of PCC and MPFC in the aMCI group is in line with a previous study, which reported that MPFC and PCC are two main nodes of the DMN [43]. Moreover, we can conclude that PCC is a powerful hub, defined as a structural and functional core across the whole brain [43,51]. Brain deposition of β-amyloid [52,53], cortical thinning and FC dysfunction [54,55,56], and a pattern of hypo-metabolism in FDG-PET studies [53] in DMN structures in AD patients from preclinical stages could explain the disruption of EC and new signaling pathways in the aMCI and AD groups.

In 2019, Ye et al. studied 30 AD, 14 MCI, and 18 normal aging subjects and performed GCA analysis on nodes of DMN, SN, and ENC networks with DPARSF and SPM8 software. In comparing AD and normal, they found decreased input to PCC, increased output from PCC (we also found the same path in our study of DMN comparing AD and normal), and decreased output from dorsal ACC (we found increased output from ACC). Furthermore, comparing MCI and normal, there was increased output from ACC (our study on the SN network of the MCI group compared to the normal group confirmed the increase of the ACC output) and inhibitory connection from PCC to hippocampal formation and from the thalamus to PCC [3].

Zhong et al. 2014 performed ICA and GCA to investigate DMN components and EC patterns; they found a decrease in the intensity and quantity of connections in AD compared to the normal group. In the normal group, the PCC was the center of convergence and received interactions from other regions. The interaction between MPFC and bilateral IPC in the normal group was weaker than in patients with AD. In our study, we also saw a decrease in the strength and intensity of connections in the AD group compared to the normal group, but the selected nodes were somewhat different, and our analysis method was spDCM [27].

In the CN of AD versus aMCI and normal group, inter-regional causal interactions increased from cerebellar anterior to cerebellar posterior, not vice versa. This situation probably reflects the network’s efforts to prevent segregation following a connection disruption.

In DAN, the FEFR→FEFL signaling pathway in all three groups was preserved. This pathway probably plays a vital role in maintaining network integrity, and FEFR and FEFL may have a driving role in the DAN. Signal cut-off in IPS(R)→FEFR and IPS(R)→IPS(L) connections and a new signaling path from IPS(L)→FEFL can distinguish the AD group from the normal group. Our findings show that FEFR could be considered a hub in normal and aMCI groups due to the + and – causal interactions with the origin and destination of this node. In another study, it was found that in the MCI group, the FEFR was involved during the encoding memory task instead of the FPN in the normal group, indicating a loss of top–down control and impaired memory and learning in this group [57]. Studies have suggested that DAN, which is responsible for the preparation and selection of responses, shows decreased intrinsic activity during rest in AD compared to the normal group [32,58]. In line with our findings, several studies show that in DAN, the severity of disconnection increases with progressing disease [14]. Neuroimaging and behavioral studies show significant impairment in the function and structures of the dorsal attention systems [14]. Moreover, several studies have shown a decrease in FC in the DAN of AD patients [14,15], which could explain the disruption of the EC and the signaling pathway of this network in the AD group.

SMN survey shows a bi-directional causal interaction in the normal group in SML(L)⇋SML(R), which has become one-sided in aMCI and AD groups from SML(R) to SML(L). Furthermore, a new connection has been established in aMCI from SML(R) to SMS. Through these connections, it is possible to differentiate between aMCI and the normal group via SMN surveys. Some task-related fMRI studies have shown decreased premotor cortex activation in the SMN of AD patients [28,54,59]. Analysis of rs-fMRI data also showed significant functional abnormalities in the SMN of patients with AD. It is probably for this reason that patients with AD exhibit subtle motor impairment [54,60] and have a signal disruption in this network.

In the VN of the aMCI group, there is a bilateral connection between VLR and VL. In the normal group, this direction is only from VLL to VLR, and in the AD group from VLL to VLR. Therefore, this bidirectional connection in the aMCI group is an indicator to distinguish aMCI from the other two groups. The VM→VO causal interaction is maintained in three groups with good strength. It seems that this connection with its driving role is an essential pathway in the integrity of the network. Although the causal interaction of VM→VO has been preserved in the three groups, this connection is weaker in the aMCI group than the normal group and attenuated in AD compared to aMCI. This finding is consistent with previous studies; a reduction in the inhibitory effects of top–down connections was found in early AD, which was associated with functional deactivation in primary visual areas [52,61]. With the progression of the disease, EC is weakened more on the left side than on the right, which is consistent with a survey that showed that all intrinsic connections, especially in the left hemisphere, were significantly weaker [52].

A Survey of the LN showed that compared to the normal group, a general impairment is observed in causal influence in the form of an increase and decrease in EC strength and a reversal of IFGR⇄IFGL signal direction in the aMCI group. The signaling map of this network became more complex in the AD group. FC dysfunction in AD patients’ networks [54] could explain the EF dysfunction in our findings. Hence, FC alteration in AD’s LN, which is associated with white matter fiber damage [62], could explain the complexity of the LN of AD patients.

In SN of the aMCI group, we see overall attenuated causal interaction in connections, inversion of SMGR⇄AAL and RPFCR⇄RPFCL connections, and disruption of information flow in these paths. Our findings showed the complexity of the SN with more EC strength in the AD group. Neuroimaging and behavioral findings show significant damage in AD patients’ ventral attention system functional structures [14], which corresponds to the dysfunction and complexity of signaling in this network. The role of the SN has been characterized as a functional sub-network in top–down control of attention [10], and increased connectivity and network complexity can disrupt this function. SN has a role in dynamic switching between the rest and task state to preserve a co-activation framework named the triple network. In AD, the underlying signal in the triple network could be inferred as an improper shift of attention between external and internal events.

In FPN, from the normal group to AD, the causal influence was altered. We also found that PPC(L) could be considered a hub in the normal group due to the robust + and – causal interactions with the origin and destination of this node. Several PET studies have shown increased deposition of Aβ in AD patients’ temporal, frontal, and parietal cortexes [63]. The decrease in FC and disconnection process in both the frontal and parietal areas might be structurally related to disconnection followed by Aβ deposition [52,63]. These neurodegenerative processes can cause the observed disorder in causal interactions of this network. FPN is the well-established functional subnetwork in top–down attention control [10]. In line with our finding, in another study, a reduction in frontal–parietal pathways (from frontal to parietal cortex) was found, which was interpreted due to gray matter volume loss in that region (resulting in impaired top–down attentional control in early AD) [63]. 

The present study has limitations in the characteristics of the cohort and methodology. The number of men and women was unequal, and the sample size was not large. The significance of the cerebral map of the three groups requires further study with a larger sample size. We performed a spDCM analysis that was restricted to the selected ROIs. In future work, imaging-based biomarkers could probably be achieved through a comprehensive study with a large sample size, artificial intelligence or machine learning method, and simultaneous utilization of the Granger causal modeling and spDCM analysis to investigate causality within–between networks.

## 5. Conclusions

Identification of directional brain interactions (extracted from the rs-fMRI method) can help to understand events related to Alzheimer’s disease. Cognitive and neuropsychic changes in AD and aMCI patients are associated with the underlying pathological spread and its effect on the signal pathway. We found impaired information flow, and disrupted causal interaction in the resting-state networks of aMCI and AD were found versus normal groups. Therefore, abnormal EC may be an indicator of cognitive decline in patients with aMCI and late-onset AD. The findings highlight the importance of EC changes in the pathophysiology of aMCI and late-onset AD. Simultaneous study of causal interactions and investigation of accumulation of neuropathology with large sample sizes in the brain’s resting state networks could help us understand the direction and strength of a signal related to the deposition of AD pathological markers.

## Figures and Tables

**Figure 1 brainsci-13-00265-f001:**
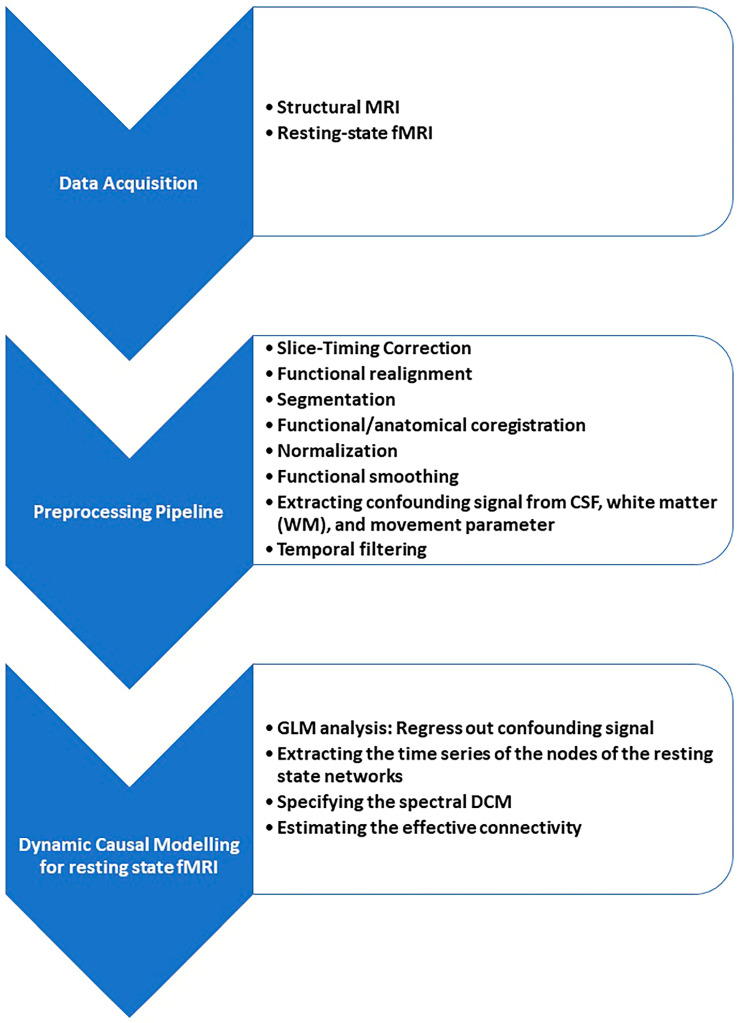
Processing the MRI/fMRI, specifying Dynamic Causal Modeling (DCM), and estimating the effective connectivity.

**Figure 2 brainsci-13-00265-f002:**
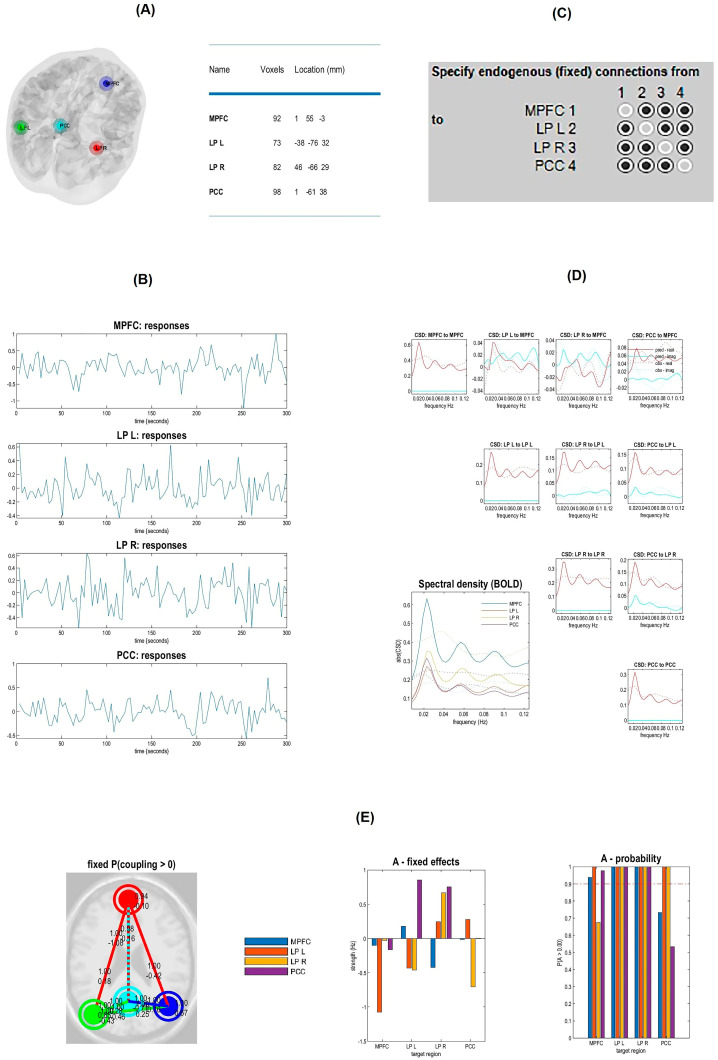
Spectral dynamic causal modeling analysis in the default mode network (DMN). (**A**) The coordinates’ nodes (MPFC: medial prefrontal cortex, LP(L): left lateral parietal, LP(R) right lateral parietal, PCC: posterior cingulate cortex) within the DMN. (**B**) Extracting Time series of DMN ‘s nodes. (**C**) Choosing model parameters (DCM with a fully connected model with no exogenous inputs and non-stochastic effect). (**D**) The plot of predicted (solid lines) and observed (dotted lines) cross-spectral densities after convergence show both self and cross spectra for four nodes. The lower panel describes self-cross spectra for four nodes. (**E**) The result of effective (fixed) connectivity between DMN ‘s nodes. The first and second panels show the estimated fixed A matrix, and the third panel shows the posterior probabilities of these endogenous connectivity parameters. The dashed line with red color indicates the 95% threshold.

**Figure 3 brainsci-13-00265-f003:**
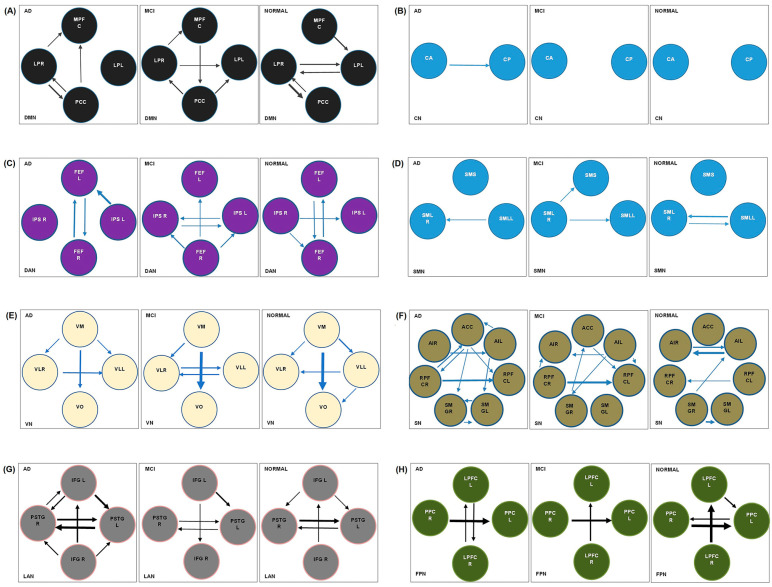
Effective connectivity patterns of AD, aMCI, and normal group in resting-state networks extracted from the spectral DCM analysis. The arrow’s direction shows the direction of causal influences, and its thickness shows the strength of the connection. The circle refers to the nodes of each resting-state network. Only connection strengths greater than 0.05 Hz are displayed (see Appendix A): (**A**) default mode network: medial prefrontal cortex (MPFC), left lateral parietal (LP(L)), right lateral parietal (LP(R)), and posterior cingulate cortex (PCC); (**B**) cerebellar network: anterior cerebellar (CA), posterior cerebellar (CP); (**C**) dorsal attention network: left frontal eye field (FEFL), right frontal eye field (FEFR), left intraparietal sulcus (IPS(L)), right intraparietal sulcus (IPS(R)); (**D**) sensorimotor network: Left lateral sensorimotor (SML(L)), right lateral sensorimotor (SML(R)), superior sensorimotor (SMS); (**E**) visual network: visual medial (VM), visual occipital (VO), left lateral visual (VLL), and right lateral visual (VLR); (**F**) salience network: anterior cingulate cortex (ACC), left anterior insula (AlL), right anterior insula (AIR), left rostral prefrontal cortex (RPFCL), right rostral prefrontal cortex (RPFCR), left supramarginal gyrus (SMGL), right supramarginal gyrus (SMGR); (**G**) language network: left inferior frontal gyrus (IFGL), right inferior frontal gyrus (IFGR), left posterior superior temporal gyrus (pSTGL), right posterior superior temporal gyrus (pSTGR) (**H**) frontoparietal network: Left lateral prefrontal cortex (LPFCL), left posterior parietal cortex (PPCL), right lateral prefrontal cortex (LPFCR), right posterior parietal cortex (PPCR).

**Table 1 brainsci-13-00265-t001:** Demographic and cognitive information.

	Late-Onset AD	aMCI	Normal	*p* Value	Post Hoc Pairwise Comparisons	Observed Power
	(*n* = 13)	(*n* = 16)	(*n* = 14)			
Age (years)	77.77 ± 7.95	72.44 ± 7.11	71.57 ± 7.14	0.062	-	0.696
Gender (female–male)	5–8	6–10	6–8	0.95	-	0.88
FAST	4–5	3	1	<0.001	Normal vs. aMCI: < 0.001	0.73
					Normal vs. AD: < 0.001	
					aMCI vs. AD: < 0.001	
MMSE	18.62 ± 1.39	25.06 ± 2.24	28.07 ± 0.83	<0.001	Normal vs. aMCI: 0.031	0.99
					Normal vs. AD: < 0.008	
					aMCI vs. AD: < 0.001	
MoCA	16.15 ± 3.53	22.19 ± 3.94	27.29 ± 1.14	<0.001	Normal vs. aMCI: 0.003	0.98
					Normal vs. AD: < 0.001	
					aMCI vs. AD: < 0.001	

*p-level* < 0.05. There was no significant difference in age among the three groups. Alzheimer’s disease after the age of 60 (Late-onset AD). Amnestic mild cognitive impairment (aMCI). Functional Assessment Staging (FAST). Mini-Mental State Examination (MMSE). Montreal Cognitive Assessment test (MoCA).

## Data Availability

The data presented in this study are available on request from the corresponding author. The data are not publicly available due to the QMISG restrictions (https://qmisg.com/ (accessed on 23 September 2021).

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
