# Peer review of "Effective Connectivity Evaluation of Resting-State Brain Networks in Alzheimer’s Disease, Amnestic Mild Cognitive Impairment, and Normal Aging: An Exploratory Study"

_brainsci, 2023, doi:10.3390/brainsci13020265_

Round 1
Reviewer 1 Report
1. The title should mention the article type.
2. Inclusions & exclusion criteria. How were assessed possible confounding medications in use by the individuals? Smoking and use of substance abuse?
3. Could the authors do a figure of the methodological flow? It is difficult to understand the methodology of the manuscript.
4. A specific chapter describing statistics should be done. How was calculated the power of the study? How were confounding variables assessed? How were the variables distributed?
Author Response
Author response: Thank you for pointing out these valuable comments and excellent suggestions.
Response to each comment is provided in the following lines:
Comment 1( Reviewer 1): The title should mention the article type.
Author response: We think this is an excellent suggestion, and to address this comment, we modified the title on pages 1, lines 2-5.
Comment 2 (Reviewer 1): Inclusions & exclusion criteria. How were assessed possible confounding medications in use by the individuals? Smoking and use of substance abuse?
Author response: To address this comment, we modified the text and added a new “Inclusion criteria” title on page 4, lines 134-147. Exclusion criteria and substance abuse have already been mentioned in the main manuscript in lines 149-157.
Comment 3 (Reviewer 1): Could the authors do a figure of the methodological flow? It is difficult to understand the methodology of the manuscript.
Author response: Thank you for this suggestion. We added figures 1 and 2 and figure legends 1 and 2 on pages 6-7.
Comment 4 (Reviewer 1): A specific chapter describing statistics should be done. How was calculated the power of the study? How were confounding variables assessed? How were the variables distributed?
Author response: To address this comment, we modified the text and added Table 1 on page 8 and a new chapter entitled” Statistical Analysis” on page 8, lines 245-260. Also, we have added additional sentences on page 5 and lines 203-212.

Reviewer 2 Report
Dear authors thank you very much for the opportunity to review this interesting manuscript. The manuscript discusses a very interesting topic. My major consideration for this work is the small sample that is participating in this study.
2. The Introduction must be enhanced. You must present new studies that are associated with your hypothesis.
3. In the second paragraph in line 54 you mention that "In the first stage, neuropsychological tests are helpful, but sensitive biomarkers to 54 pathological changes could be more predictive and Informative" You have to clarify this sentence. That means that all the other studies that have discussed MCI or Alzheimer's diseases have major methodological issues. Present references and studies that support this specific sentence. If not I suggest withdrawing it.
3. You have to present your hypothesis with references that support your experimental design.
4. Before presenting your results you have to write a paragraph explaining the analysis that you applied.
5. More clarification is needed in the Conclusions.
Author Response
Author response: Thank you for pointing out these valuable comments and excellent suggestions.
Response to each comment is provided in the following lines:
Comment 1 (Reviewer 2): The Introduction must be enhanced. You must present new studies that are associated with your hypothesis.
Author response: To address this comment, we modified the text on pages 2-3, lines 58-66 and 82-114.
Comment 2 (Reviewer 2): In the second paragraph, in line 54 you mention that "In the first stage, neuropsychological tests are helpful, but sensitive biomarkers to 54 pathological changes could be more predictive and Informative" You have to clarify this sentence. That means that all the other studies that have discussed MCI or Alzheimer's diseases have major methodological issues. Present references and studies that support this specific sentence. If not, I suggest withdrawing it.
Author response: Thank you for this suggestion. This sentence is now removed from the manuscript.
Comment 3 (Reviewer 2): You have to present your hypothesis with references that support your experimental design
Author response: As suggested by the reviewer, we presented our hypothesis on page 3 in lines 115-120, and we added references that support our experimental design on pages 2-3 and lines 94-114.
Comment 4 (Reviewer 2): Before presenting your results, you have to write a paragraph explaining the analysis that you applied.
Author response: According to your valuable suggestion, before presenting results, we write a paragraph explaining the analysis on page 8, lines 245-260.
Comment 5 (Reviewer 2): More clarification is needed in the Conclusions.
Author response: Thank you for pointing this out. We modified the text on pages 13, lines 470-473 and 477-480.

Reviewer 3 Report
Introduction
- The Introduction section needs to be more complete. Please include previous studies about connectivity and resting-state brain networks in patients with Alzheimer's disease (AD) or mild cognitive impairment, which could provide this study's current background.
-We suggest including a hypothesis for this study even when this is exploratory
Methods
-Please describe whether some participants suffered from chronic diseases such as hypertension or diabetes. Do you think that this might affect your results?
Results
-Please report some of the statistical measures in the manuscript; this may clearer the results
-I suggest avoiding subjective descriptions of brain networks, such as "connection is weakened" or "connection is almost disconnected." Please explain what kind of statistical arguments you must describe these differences between groups.
-The tables note need to be completed. Please verify that you are defining each acronym.
Discussion
-I suggest moving multiple paragraphs from the discussion to the Introduction section, then you can state a hypothesis and discuss it.
Example 1 in your manuscript: "There are several known features to 241 identify AD: global atrophy and local atrophy (prominent atrophy of the medial temporal 242 lobe) [25], β-amyloid deposition in dysfunctional hub centers [26], hypometabolism in the 243 temporal, parietal, and posterior cingula [25], disruption of EC in the resting-state net- 244 work of AD and aMCI can be due to these disorders. As our study showed, another study 245 also reported a decrease in EC in strength and quantity in AD compared to the normal 246 group [27]."
Example 2: "Several studies have shown a decrement in volumetric markers 321 [47, 48] the presence of amyloid markers ]49 ,48[, and a decrease in the FC of AD and 322 aMCI patients' SN [40, 41]"
Example 3: "The SN, involved in auto- 325 nomic and homeostatic processing, has been found to show increased connectivity in AD 326 patients [30, 50]. In the AD group, decreased connectivity within the DMN is often 327 Brain Sci. 2023, 13, x FOR PEER REVIEW 8 of 17 accompanied by increased connectivity in the salient network, possibly due to compensa- 328 tory mechanisms [51, 52].
Author Response
Author response: Thank you for pointing out these valuable comments and excellent suggestions.
Response to each comment is provided in the following lines:
Comment 1 (Reviewer 3): The Introduction section needs to be more complete. Please include previous studies about connectivity and resting-state brain networks in patients with Alzheimer's disease (AD) or mild cognitive impairment, which could provide this study's current background.
Author response: To address this comment, we modified the text on pages 2-3, lines 58-66 and 82-114.
Comment 2 (Reviewer 3): We suggest including a hypothesis for this study even when this is exploratory.
As suggested by the reviewer, we presented our hypothesis on page 3 in lines 115-120.
Comment 3 (Reviewer 3): Please describe whether some participants suffered from chronic diseases such as hypertension or diabetes. Do you think that this might affect your results?
Author response: To address this comment, we modified the text and added a new title, “Inclusion criteria” on page 4, lines 134-147. Exclusion criteria and substance abuse have already been mentioned in the main manuscript in lines 149-157. Hypertension and diabetes are our exclusion criteria.
Comment 4 (Reviewer 3): Please report some of the statistical measures in the manuscript; this may clearer the results.
Author response: To address this comment, we modified the text and added Table 1 on page 8 and a new chapter entitled” Statistical Analysis” on page 8, lines 245-260. Also, we have added additional sentences on page 5 and lines 203-212. Also, we have added a paragraph on the result section on page 8 lines 262-268.
Comment 5 (Reviewer 3): I suggest avoiding subjective descriptions of brain networks, such as "connection is weakened" or "connection is almost disconnected." Please explain what kind of statistical arguments you must describe these differences between groups.
Author response: Thank you for this suggestion. As all groups were analyzed, tested, and reported in the same way, we directly compared the amount of significant results. Tables S2-S8 are the statistical description of the networks.
Comment 6 (Reviewer 3): The tables note need to be completed. Please verify that you are defining each acronym.
Author response: To address this comment, we modified the tables note.
Comment 7 (Reviewer 3): I suggest moving multiple paragraphs from the discussion to the Introduction section, then you can state a hypothesis and discuss it.
Example 1 in your manuscript: "There are several known features to 241 identify AD: global atrophy and local atrophy (prominent atrophy of the medial temporal 242 lobe) [25], β-amyloid deposition in dysfunctional hub centers [26], hypometabolism in the 243 temporal, parietal, and posterior cingula [25], disruption of EC in the resting-state net- 244 work of AD and aMCI can be due to these disorders. As our study showed, another study 245 also reported a decrease in EC in strength and quantity in AD compared to the normal 246 group [27]."
Example 2: "Several studies have shown a decrement in volumetric markers 321 [47, 48] the presence of amyloid markers ]49 ,48[, and a decrease in the FC of AD and 322 aMCI patients' SN [40, 41]"
Example 3: "The SN, involved in auto- 325 nomic and homeostatic processing, has been found to show increased connectivity in AD 326 patients [30, 50]. In the AD group, decreased connectivity within the DMN is often 327 Brain Sci. 2023, 13, x FOR PEER REVIEW 8 of 17 accompanied by increased connectivity in the salient network, possibly due to compensa- 328 tory mechanisms [51, 52].
Author response: We think this is an excellent suggestion, and to address this comment, we modified the text and moved these paragraphs from the discussion to the Introduction section. Pages 2-3, lines 61-66 and 102-105.

Reviewer 4 Report
Dear Authors,
first of all, I thank You for giving me the opportunity to read this Your manuscript, submitted for publication in Brain Sciences.
I have only minor comments.
1. Consider a comparison between your study and other similar studies in published literature;
2. Please, report the age of the patients in the three groups You assessed;
3. References should be updated. For instance, Dennis EL, Thompson PM. doi: 10.1007/sl1065-014-9249-6 should be added;
4. Gramatical and stylistic revision would be useful.
Author Response
Author response: Thank you for pointing out these valuable comments and excellent suggestions.
Response to each comment is provided in the following lines:
Comment 1 (Reviewer 4): Consider a comparison between your study and other similar studies in published literature.
Author response: To address this comment, we modified the text, and we have added additional sentences on pages 2-3 and lines 94-114, and page 11, lines 371-387.
Comment 2 (Reviewer 4): Please, report the age of the patients in the three groups You assessed;
Author response: To address this comment, we modified the text and added Table 1 on page 8.
Comment 3 (Reviewer 4): References should be updated. For instance, Dennis EL, Thompson PM. Doi: 10.1007/sl1065-014-9249-6 should be added;
Author response: As suggested by the reviewer, we updated the references.
Comment 4 (Reviewer 4): Grammatical and stylistic revision would be useful.
Author response: We have revised the grammar and style according to your valuable suggestion.

Round 2
Reviewer 2 Report
The manuscript is revised according to the comments and quidelines
Reviewer 3 Report
Congratulations, well done :).
Reviewer 4 Report
Dear Authors,
all my comments and suggestions were satisfactorily met in the revised version of Your manuscript.
My overall recommendation is that it can be accepted for publication.